# Influence of Information Blocking on the Spread of Virus in Multilayer Networks

**DOI:** 10.3390/e25020231

**Published:** 2023-01-27

**Authors:** Paulina Wątroba, Piotr Bródka

**Affiliations:** 1Capgemini, Legnicka 48K, 54-202 Wrocław, Poland; 2Department of Artificial Intelligence, Faculty of Information and Communication Technology, Wrocław University of Science and Technology, 50-370 Wrocław, Poland

**Keywords:** coexisting spreading processes, epidemics, network science, multilayer networks

## Abstract

In this paper, we present the model of the interaction between the spread of disease and the spread of information about the disease in multilayer networks. Next, based on the characteristics of the SARS-CoV-2 virus pandemic, we evaluated the influence of information blocking on the virus spread. Our results show that blocking the spread of information affects the speed at which the epidemic peak appears in our society, and affects the number of infected individuals.

## 1. Introduction

The increases in human mobility and globalization have created ideal conditions for the spread of new epidemics [1]. At the turn of 2019 and 2020, a new coronavirus started spreading in Wuhan. According to the University of Toronto Citizen Lab report [2], information about it was not released to the general public for more than three weeks, and multiple other sources report that there was an active campaign to limit the spread of information about the virus [3,4,5,6,7,8,9,10]. Since spreading the information (awareness that there is a virus circulating in society) is an important tool in limiting the spread of the virus [11,12,13,14,15,16,17] (aware people might take preventive actions such as staying at home, wearing face masks, washing hands more often, etc.), we asked questions regarding how delaying information spread influenced the spread of the virus, and how it affected the number of infected individuals and disease dynamic.

Unfortunately, we were not able to find the answers to those questions in the related works; thus, we developed a model for the interaction between virus and information spreading in multilayer networks (Section 2), where becoming aware of the virus results in limiting the chance of becoming infected. Next, we adjusted the model using the COVID-19 pandemic data from its early days (Section 2.3). Finally, we performed experiments (Section 3) to analyze and compare the spread of SARS-CoV-2 in three scenarios: (i) only the virus spreads, (ii) the virus and information spread simultaneously from the beginning, and (iii) the virus and information spread simultaneously, but the information spread is delayed for some period of time.

## 2. Materials and Methods

In this section, we briefly introduce the most important concepts and assumptions for our experimental part.

### 2.1. Multilayer Network

To evaluate our ideas in a more realistic scenario, we decided to use the multilayer network [17,18,19,20], where the network is defined as M=(N,L,V,E) [20], where

*N* is a not-empty set of actors $\left\{n_{1},...,n_{n}\right\}$;*L* is a not-empty set of layers $\left\{l_{1},...,l_{l}\right\}$;*V* is a not-empty set of nodes, V⊆N×L;*E* is a set of edges $\left(v_{1},v_{2}\right):v_{1},v_{2}\in V$, and if $v_{1}=\left(n_{1},l_{1}\right)$ and $v_{2}=\left(n_{2},l_{2}\right)\in E$, then $l_{1}=l_{2}$.

The example of a multilayer network is presented in Figure 1. This network contains:Six actors $\left\{n_{1},n_{2},n_{3},n_{4},n_{5},n_{6}\right\}$;Two layers $\left\{l_{1},l_{2}\right\}$;Ten nodes $\{v_{1}=\left(n_{1},l_{1}\right),v_{2}=\left(n_{2},l_{1}\right),v_{3}=\left(n_{3},l_{1}\right),v_{4}=\left(n_{4},l_{1}\right),v_{5}=\left(n_{5},l_{1}\right),$$v_{6}=\left(n_{1},l_{2}\right),v_{7}=\left(n_{2},l_{2}\right),v_{8}=\left(n_{3},l_{2}\right),v_{9}=\left(n_{4},l_{2}\right),v_{10}=\left(n_{6},l_{2}\right)\}$;Eleven edges $\{\left(v_{1},v_{2}\right),\left(v_{1},v_{5}\right),\left(v_{2},v_{5}\right),\left(v_{2},v_{3}\right),\left(v_{2},v_{4}\right),\left(v_{3},v_{4}\right),\left(v_{6},v_{9}\right),\left(v_{6},v_{10}\right),$$\left(v_{7},v_{8}\right),\left(v_{7},v_{9}\right),\left(v_{8},v_{9}\right)\}$.

This network model allows us to have two different networks (layers), the first one for disease spreading, which for obvious reasons needs to be limited to offline world contacts to support virus spread, and the second one for “online” contacts that allow information spreading. Both layers can have a completely different topology, e.g., two people living in two geographically distant cities may never meet, but they can exchange information via phone or social platforms; on the other hand, two people can exchange viruses because they have shared the same shopping cart or used the same bus, but they might never talk and exchange information.

### 2.2. Spreading Models

#### 2.2.1. Epidemic Spreading

In the SIR model, every person who belongs to a population, also called an actor or a node, can be in one of three states: *S* (susceptible), in which a person is susceptible to infection; *I* (infected), which means infected and at the same time spreading the disease, and *R* (recovered), when a person has recovered and acquired immunity or has died and can no longer infect or become sick again (e.g., smallpox, mumps, and other diseases for which people can be vaccinated).

A susceptible actor can be infected by an infected actor in one cycle with probability β, while infected actors can recover in each cycle with probability γ. This process can be described by the following equations:$$\frac{ds}{dt}=-\beta is,\;\frac{di}{dt}=\beta is-\gamma i,\;\frac{dr}{dt}=\gamma i,$$
where i,s,r represent the fraction of susceptible, infected, and recovered individuals in the total population, respectively. The state changes are also presented in Figure 2.

It must be noted that in a real epidemic spreading, for diseases such as chickenpox or mumps, a person in state *S* will be infected only if he or she has direct contact with an infected person.

Nevertheless, in a complex network, actors are represented by nodes, and the possibility of contact is determined by connections between them, i.e., edges in the network. In such circumstances, a node in state *S* may change its state to *I* only if it has at least one infected neighbor. In this way, classical epidemic models can be extended to network representation, and the presented expressions can be considered as a special case where the corresponding network is fully connected. In the absence of some connections in the network, the fraction of susceptible individuals in the total population may be larger, and there may be actors who will not be infected [21].

#### 2.2.2. Information Spreading

In the SIS model, an actor can be in one of two states: susceptible (*S*) or infected (*I*). The person’s state change is determined by the relevant probability. If an actor is in the *S* state, its switch to *I*, in any iteration, will depend on probability β. Return from the state *I* to *S* depends on the probability γ. This reflects the situation where a susceptible person becomes infected by any infected member of the population and then becomes ill but has not acquired immunity. It means that despite being already ill, a person is susceptible to reinfection (e.g., cold or seasonal flu). These processes can be described by the following equations:$$\frac{ds}{dt}=-\beta is+\gamma i,\;\frac{di}{dt}=\beta is-\gamma i,\;$$
where i,s represent the fraction of susceptible and infected actors in the total population, respectively. The state changes are also presented in Figure 3.

This model corresponds to real processes of spreading seasonal diseases such as cold or flu, for which one does not acquire immunity, as in the case of chickenpox or mumps. The representation of this model for the network is similar to the case described for the SIR model, except that instead of a transition to state *R*, there will be a return to state *S* [21].

Epidemic models can be used to simulate other spreading processes. The most common example would be information spreading, where we have models such as UAU [22] (unaware–aware–unaware) or UAF [23] (unaware–aware–forgot), which are based on SIS and SIR models, respectively. In the case of our research, we decided to use an SIS-based model (i.e., UAU [22]) to model the spread of information.

#### 2.2.3. Interaction between Processes

Interactions between multiple processes in a network can take many forms, and most research is centered on one of three categories: supporting, competing, and mixed approaches [17].

**Supporting processes** are observed, for example, in the case of opinion formation and decision-making, where public opinion about a topic is taken into account during the decision-making process [24].

Epidemics in multilayer networks can take a cooperative form, as one disease can exacerbate or inhibit the development of another [25]. As a result, the dynamics and extent of a disease can be increased by other diseases spreading in the same network. One disease may be a consequence of contracting another. For example, the number of people with tuberculosis increases in the population with HIV [26]. However, the issue of mutual support processes represents a small percentage of all works [17].

**Competing processes**. Competition between processes has been modeled and analyzed on a large scale. An example can be competition studies for memes [27] and extended to generalization for other content [28].

Competitive processes have also been studied in the context of optimal resource allocation in multilayer networks, where a single node may participate in multiple processes at the same time. It has also been shown that the diffusion of resources in the information layer can affect the spread of outbreaks in the physical contact layer and change the phase transition. Studies have shown that the existence of optimal resource diffusion leads to maximum disease suppression [29].

**Mixed approaches**. The third type of interaction is a mixed approach, used in modeling competing and supporting processes spreading simultaneously. For example, the appearance on the market of new technologically advanced products, which are very similar to each other, creates a demand for new services (support) and, at the same time, strengthens the competition on the market (competition) [30].

Researchers also analyzed the coexistence of cooperation and competition mechanisms. They observed that increased cooperation boosts the ability of content to spread across all layers, whereas without cooperation, the layers are independent, and each virus spreads only within one layer. Due to the competition mechanism, only one viral agent can be assigned to one node [31].

Interesting results can be observed in the field of spreading diseases and information about them. In some research, awareness inhibits the spread of diseases. On the other hand, we can have a situation in which the infected node becomes aware and can spread infection and information about the disease at the same time [12,15,16]. A similar scenario in which disease supports the spread of information, and awareness reduces disease, was also examined for disease and immunization. The spread in a multilayer structure that contains disease and immunization can enhance or dampen the epidemic. While immunization can compete with the epidemic, it can also enhance its dynamics [32]. Mixed interactions can also be observed in individuals waiting for immunization [33].

### 2.3. Adjusting Parameters for COVID-19 Pandemic

Based on previous research in epidemic modeling, we adjusted the model’s parameters to the SARS-CoV-2 virus and the early days of the COVID-19 pandemic. Out of many existing epidemic models, for virus spread, the SIR model was selected, and for information spread, the UAU (SIS) model was chosen. Values for all parameters can be found in Table 1 and the explanation for those values can be found in the following subsections.

#### 2.3.1. SIR Model

In the beginning, it was necessary to define the initial conditions and assumptions resulting from the specificity of the virus, as well as the research questions posed. That was equal to answering the questions *where?*, *what?*, and *how?* it spreads.

**Spreading structure.** To simulate the coronavirus epidemic, it was necessary to select the structure in which it would occur. In the real world, the virus is spread through direct contact between a susceptible person and an infected person or through things/objects/surfaces upon which virus particles have settled. Additionally, the situation is complicated because susceptibility to infection is an individual factor. Moreover, in the case of the analyzed virus, this factor is greater for the elderly or people suffering from chronic diseases.

**Initial state.** An essential question is *how to initiate an outbreak*. Based on the literature analysis, there are two main approaches to establishing the initial state of the network. In the first one, the epidemic starts with the disease of a certain number of individuals, the most common being the so-called patient zero. This approach, as faithful as possible to the principles of epidemiology, is slightly troublesome from the perspective of comparative studies when we test networks of different sizes [34]. An alternative approach is more sympathetic to comparative analysis, as it assumes that some percentage of nodes in a network or layer is initially infected [35]. Due to the prospect of comparing epidemic progression for networks of different sizes, we used the second approach.

In the initial state, one percent of all nodes in the personal contact layer are infected. The calculated number of infected actors is rounded up to an integer value to ensure that for networks with the number of nodes in the contact layer below 100, we have at least a single seed node. Determining the actors who will be infected is the result of random selection.

**State changes.** An actor in state *S* can change its state with probability β to *I* if it has an infected neighbor. This model means that for each actor in the state *I*, all direct neighbors (all nodes connected by an edge to a given infected node) are searched; for each of them, a value between (0, 1) indicating the probability of infection is randomized that is compared with the value of the threshold β. If the drawn value is less than the β then this actor will change its state to *I* on the next epidemic day (iteration of the process). Otherwise, the state of the node will not change. A separate draw determines the change of state of an actor in the state *I*. When all neighbors of the infected individual are found, a probability value is generated from the interval (0, 1) for it and, similarly to the case described above, it is compared with γ. The change in state to *R* will occur only if the generated value is lower than the γ. If this does not happen, the actor remains in the state *I* and continues to infect.

It should be noted that a node that has changed its state to *R* cannot change it to *I* again. However, there are indeed reports of reinfection in the literature, but their percentage relative to all cases is so low that they are not included in this model.

**Probabilities β and γ.** The coronavirus pandemic led to intensive work in the scientific community on modeling the epidemic. As a result, in the literature, one can find probability values for the SIR model tailored to the modeling of the SARS-CoV-2 virus spreading. Most publications concern Asian countries, especially China, where the pandemic began, and European countries, where the epidemic further developed—causing paralysis of health services, resulting in serious illnesses or deaths of many people. These countries included Italy, Spain, France, and, to a lesser extent, Germany and Poland.

Based on the analysis of available works, as well as the available social networks and their density, it was decided to adopt four different probability values, the first three for Italy (β=0.19,γ=0.10 [36], β=0.22,γ=0.02 [37], β=0.28,γ=0.08 [38]) and one for Poland (β=0.31,γ=0.10 [36]).

#### 2.3.2. UAU Model

The spread of the virus is accompanied by the spread of awareness (information) of its existence. However, this is a process at least partly independent of the spread of the virus. For this reason, it is necessary to have two different models for both processes. For the spread of information about the virus, the SIS-based UAU model was adapted.

Previous research showed that despite the spreading of seasonal diseases such as cold or flu, the UAU model could be successfully adapted for the spread of different types of information, taking into account the process of forgetting [13]. The states of the model can then be described as *U* (unaware)—unaware of information (*S* in SIS model) and *A* (aware)—spreading information (*I* in SIS model).

The change of states is determined by the probabilities β and γ. To simplify the understanding of the interactions between the models, the probabilities will be denoted by symbols ϵ and μ, respectively.

In the UAU model, an unaware actor in state *U* may learn about the existence of the virus from a conscious member of the population *A*. Over time, the aware person returns to state *U*, which corresponds to the situation in which someone forgets about the existence of the virus or becomes used to it and awareness does not affect its behavior [13]; for example, someone, despite knowing about the pandemic, stops wearing the mask. Similarly to the adaptation of the SIR model, for the UAU model, it was necessary to define the initial state and the assumptions.

**Spreading structure.** Simulating the spread of information requires defining the medium in which it will occur. Information and viruses in the real world coexist within the same population. Therefore, the network for the SIS and SIR models is common. However, the specifics of spreading differ. Unlike the virus, access to information is so widespread that receiving it does not require direct contact between two people. Information reaches the recipient through social networks, the Internet, newspapers, etc. However, this does not exclude acquiring information through real interpersonal contacts or traveling by shared means of transport. Furthermore, obtaining information from one source does not prevent encountering the same information again through another medium. Therefore, to simulate a real process, information may spread throughout the network at all layers. For simplicity, the type of interaction does not affect the entire process; regardless of the layer, the assumptions are the same.

**Initial state.** Similar to the infection process, information appears in a population through human action. To determine the initial state of the aware population, the same tactics were used as for the virus. Initially, one percent of all actors in the network are aware of the virus. The selection of informed actors results from random selection similar to the SIR process.

**State changes.** An actor in state U can change its state to *A* with probability ϵ if it has an aware neighbor. What this model means is that for each actor in state *U*, all immediate neighbors (all nodes connected by an edge to a given aware vertex) are searched, and for each of them, a value is drawn from the interval (0, 1), denoting the probability of awareness. It is then compared with the value of the threshold ϵ. If the drawn value is less than the ϵ, the actor will change its state to *A* in the next iteration. Otherwise, the state of the individual will not change.

A separate drawing determines the change of state of an actor in state *A*. When all neighbors of the aware individual are found, then a probability value from the interval (0, 1) is drawn for it and, similar to the case described above, compared with the threshold, which is the probability μ. A return to state U will occur only if the drawn value is less than the μ. If this does not happen, the actor remains in state *A* and continues to spread information. A node that has changed its state to U may change it again to *A*. Although, over time, we forget a given piece of information or consciously downplay the presence of the virus, resulting in a return to state *U*, this does not preclude a renewed increase in interest or awareness.

##### Probabilities ϵ and μ

In previous research, we could not find information on how to determine the ϵ and μ for the UAU model during the spread of information about the SARS-CoV-2 virus. Therefore, it was assumed that the probabilities ϵ and μ would be equal to the probabilities of the SIR model to reflect the intensity of the spread of information and this issomehow related to the intensity of virus spread. Since in real life, the spread of information is much faster than the virus itself, it was decided to extend the set of probabilities by multiplying the initial probabilities according to the equations:ϵ=min(β∗x,1),μ=min(γ∗x,1)wherexε{1,2,3,4}.
Thus, for each β and γ combination, we have four combinations of ϵ and μ.

#### 2.3.3. Interaction between Virus and Information Processes

While analyzing the impact of the spread of information on the virus, it is necessary to locate both processes in a single medium. For this purpose, a multilayer network was chosen. The virus spread, simulated by the SIR model, will progress within a direct contact layer. In contrast, awareness will spread in all layers. Therefore, it is necessary to address the interaction between the models. In reality, awareness of the virus causes a range of behaviors designed to avoid infection (social distancing, masks, vaccination, etc.). A representation of this phenomenon will be a reduction in the infection probability, β, for actors aware of the virus. Choosing just one number for the reduction of β was difficult since different actions yield different results in infection risk reduction. For example, wearing a mask will result in 65% risk reduction (RR) [39], and one meter of social distancing has a similar effect (RR of 65%) [39,40], with RR increasing with the distance [39]. Other actions have lower (e.g., face shields) or higher RR (e.g., quarantine and self-isolation have RR of almost 100%). Additionally, one will increase RR with the combination of more than one action (e.g., face mask and social distancing). Since various countries decided on different actions and various actions have various effects we decided to assume RR of 90%. Thus, the primary probability will be reduced by a factor of ten, which the following equation can describe: $\beta^{\prime }=\frac{\beta }{10}$, where β is the probability of infection and $\beta^{\prime }$ is the probability of infection of an aware node.

Similar to how awareness affects the probability of infection, the infection can alter the chance of becoming aware. This corresponds to the situation where a person with COVID-19 becomes aware of the SARS-CoV-2 virus by having specific disease symptoms or test results. However, not all cases of infection with the coronavirus are manifested by symptoms [41,42]. At the same time, symptoms can be similar to other upper respiratory diseases that are not difficult to confuse. To address the impact of this phenomenon on the ϵ${}^{\prime }$, the percentage of symptomatic patients was taken. In previous research on the SARS-CoV-2 virus, only a few addressed the issue of the number of asymptomatic patients. One of the most important is the proportion of patients with asymptomatic COVID-19 based on observations of passengers on the Diamond Princess, a quarantined ship off the coast of Wuhan. In this case, 17.9% of the ill passengers were asymptomatic [41]. However, it should be noted that the ship’s crew, by sharing the quarantine, was an isolated community, so generalizing the results to the whole population might not be correct. Slightly more general results were obtained in a study of a group of Japanese people evacuated from Wuhan by a shared plane. Although the group examined was smaller than that of the ship’s passengers, a significant difference is the lack of shared isolation. Researchers, using a binomial distribution, estimated that among evacuees, the proportion of symptomless patients was 30.8% [42]. The characteristics of the virus change over time due to mutations or certain individual attributes in different populations. However, since we are interested in the initial part of the pandemic, we can use published data from the initial period of the spread of the SARS-CoV-2 virus. Based on this, we assumed that the probability that the unaware node becomes aware if it is infected will correspond to the percentage of symptomatically ill people from [42], i.e., $\epsilon^{\prime }=0.692$. The described changes in probabilities are presented in Table 2.

In summary, the interaction between information dissemination and virus spread can be classified as a mixed interaction. The virus spread supports information dissemination, and information dissemination can suppress virus spread. An example of support is increasing the chance of obtaining information about the virus for an infected node. This allows the information to spread faster.

Otherwise, knowledge of the existence of the virus reduces the chances of an actor being infected by blocking the development of an epidemic. This is an example of competition. It should be noted that competition is not limited to the layer where both processes occur because awareness of the virus gained in the layer of direct contacts affects the spread of information about the virus in all other layers.

## 3. Results and Discussion

Experiments were performed using *multinet library* [43] and six multilayer networks (Table 3). For each network, we selected a layer acting as a direct contact layer for virus spread. For some networks, the selection was based on the characteristic of interactions between layers (e.g., N1), while for others, the choice was arbitrary (e.g., N3). The rest of the layers acted as communication layers. For the bigger networks, N5 and N6, we ran the experiment a few times, each time with a different layer acting as the direct contact layer. We know that not all networks are classic social networks; however, we also wanted to observe the effects on other complex networks, especially since they reflect human mobility (N3) or information exchange (N5, N6).

To evaluate the relationship between information spread and virus spread, three main scenarios were constructed.

Worst case scenario: only virus spreads (SIR).Best-case scenario: virus and awareness spread simultaneously (SIR and UAU).Evaluated scenario: the virus spreads, but the information about the virus is blocked for some period. When the blocking time ends, the information (awareness) about the virus also starts to spread (blocking).

**Only virus spreads.** In the direct contact layer, one percent of all actors are infected by drawing lots. Simulations lasted 150 days, where one day is one iteration of the SIR model. During an epidemic, the probabilities β and γ are constant. The epidemic can end before 150 days if all actors are recovered or, although there are people in the susceptible state, they do not have infectious neighbors, so they cannot become infected. The first case refers to the situation in which the entire society has been infected and recovered, while the second case refers to the situation in which enough people have been infected to create herd immunity in society. For each set of parameters, the scenario was run at least 20 times.

**Virus and information spread simultaneously.** In the direct contact layer, the virus spreads the same way as when there is no information. At the same time, information about the existence of the virus spreads throughout the network. As a result of a random draw, those who are aware of the existence of the virus are selected, representing one percent of all nodes in the network. This is followed by the awareness-spreading process according to an adapted UAU model with fixed probabilities ϵ and μ. There are interactions between the processes. If an actor is aware of a virus, then their probability of being infected is changed to a smaller value $\beta^{\prime }$. In the opposite situation, for an infected actor, the probability of becoming aware is $\epsilon^{\prime }$. As in the case of the virus itself, the epidemic can end before 150 days have passed when everyone is in state *R* or the number of infected nodes is zero. The scenario is repeated at least 20 times for each combination of parameters.

**Information blocking.** Similar to the spread of a single process, the virus spreads through a layer of direct contacts. The epidemic lasted 150 days. In the initial phase of the experiment, the probabilities β and γ are constant, and only the virus spreads in the network. The spread of information begins after the blocking time, which is intended to simulate the real situation when the information about SARS-CoV-2 was not released to the public. The Citizen Lab report shows that the blocking lasted three weeks (21 days) [2]. Consequently, for the first 21 iterations, the spread of information is blocked. As networks of different sizes were analyzed, it was decided to test the effect of changing the blocking time on the outbreak and additionally test blocking for one week (7 iterations) and two weeks (14 iterations). After that time, the spread of the information begins, and the information spreads simultaneously to virus as described above (scenario: virus and information spread simultaneously). The epidemic lasts 150 days or until all nodes are in the *R* state or no actor is in the *I* state. The scenario is repeated at least 20 times for each combination of parameters.

### 3.1. Effect of Information Blocking

To compare three scenarios with each other, we looked at three moments during the epidemic spreading:When the peak of the infected people occurred, assuming that the later this happened the better, as it gives the healthcare services more time to prepare. Thus, we took the peak day for case 2 (virus and information spread simultaneously) and calculated how much faster we would have the peak day for the other two cases.How many people became infected till the peak day? We took the peak day for the second case (virus and information) and compared how many more people became infected until this day in the other two cases (taking into account both infected and recovered nodes).How many people became infected during 150 days? We took the final number of infected and recovered people for the second case (virus and information) and compared how many more people became infected during 150 days in the other two cases (taking into account both infected and recovered nodes).

Table 4 and Figure 4 present the summary of our results for the three aspects mentioned above. We can see that in most networks, the results indicate that blocking information for just 21 days results in the peak day being up to 35% (network N2) faster than in the case where the information can spread together with the virus. A similar case occurs with the number of infected individuals on the peak day, which can be up to 138% (again for network N2) higher than for the SIR and UAU process. Interestingly, while information blocking significantly impacts the “peak time”, it has a lower impact on the total number of individuals affected by the disease after 150 days.

We have to note that since both SIR and UAU are not deterministic processes, we repeated the simulation at least 20 times for each combination of parameters. However, for bigger networks such as N6-dis-nn, N6-MN, and N6-SI, this number was too low, and in the case of those networks, all three cases (SIR, SIR, and UAU, blocking) were very similar. Each one was within the standard deviation of another two, and there was no statistically significant difference between all three cases (Table 5). Unfortunately, due to the size of the network and the number of combinations of parameters, we could not repeat the simulations more times.

### 3.2. Duration of the Delay

The next element we evaluated was the effect of the delay duration on the epidemic spreading. To do so, we ran our experiments again, this time for 7- and 14-day information blocking periods, and compared the with previous results for the 21-day blocking period. Due to the network size in this experiment, we did not use the N5 network. The results show that information blocking, regardless of the blocking period, results in very similar results, i.e., although for some individual networks, longer blocking results in a faster epidemic peak and higher number of infected nodes, on average, the results for all blocking periods are very similar (Table 6 and Figure 5), and according to Wilcoxson signed rank test, most of the differences are not statistically significant (Table 7).

This leads to the conclusion that what is important is the fact that we block information about infectious diseases, not the duration of the ban. This emphasizes the need to share information with society as soon as possible so that the information can start spreading as soon as possible and prevent as many infections as possible, especially in the first weeks of a pandemic.

## 4. Conclusions

The study included an investigation of the influence of information blocking on the spread of infectious diseases. A comparison of the intensity of the epidemic for three different periods of information blocking, as well as an investigation of the impact of the parameters of the information spreading model on the epidemic course, revealed that the spreading of information about the virus reduces the intensity of the epidemic and flattens the disease curve. No impact of shorter blocking periods on the change in epidemic dynamics was found, indicating that even a short period of information blocking will increase the size and speed of the epidemic.

### Limitations of Our Research

The problem of spreading information and virus in multilayer networks is very important. This work focused on mapping the most important features of the propagation of the SARS-CoV-2 virus and information about it. Research allowed us to investigate the most important relationships; however, some aspects must be elaborated further. In this paper, the spreading probability thresholds for the SIR model and the UAU are assumed to be the same for all actors. They only change due to interactions between processes, but each change results in the same probability value. In real life, the chance of contracting a virus is significantly influenced by age, the burden of additional diseases, and other factors. Therefore, it would be necessary to investigate how the analyzed process will shape the individual probability of infection for each actor in the network. One way of studying this dependence would be to randomize the status of individual actors in the network based on available statistics that describe the characteristics of the population of a given country through information such as gender, age, or the percentage of patients with specific diseases.

A broader view of the examined relationship between virus and information spread could be gained by extending the set of tested probabilities for the SIR model to include probability values for countries other than Italy and Poland.

The model of the spread of information about the virus could be improved with the time dependence of the probability of information spread. In this way, it would be possible to represent the real pattern that new information is more popular. Then it spreads faster in direct contacts, as well as through social networks. As the information becomes older, it becomes less popular, which means that the spread is slower, and sometimes stops completely. Additionally, our model assumes that the information spreads between actors, ignoring the external influence on the network, such as government information campaigns that target all nodes in the network simultaneously. Because of this mechanism, the influence of information blocking could be even more profound since information might reach all nodes at the beginning of the epidemic.

An additional interesting issue is an attempt to represent the phenomenon of “forgetting” or ignoring the existence of the virus. It is the result of the fatigue of having to respect the restrictions imposed by the authorities or to be careful and wear personal protective equipment. Therefore, as time passes, more and more people start to ignore the information about the existence of the virus and become less vigilant. This should be expressed as an increased chance of infection despite awareness of the virus after a certain period of the epidemic.

Finally, we used average-sized networks. It would be interesting to use larger networks that better reflex the complexity of interaction between people on various levels and consider real mobility patterns.

## Figures and Tables

**Figure 1 entropy-25-00231-f001:**
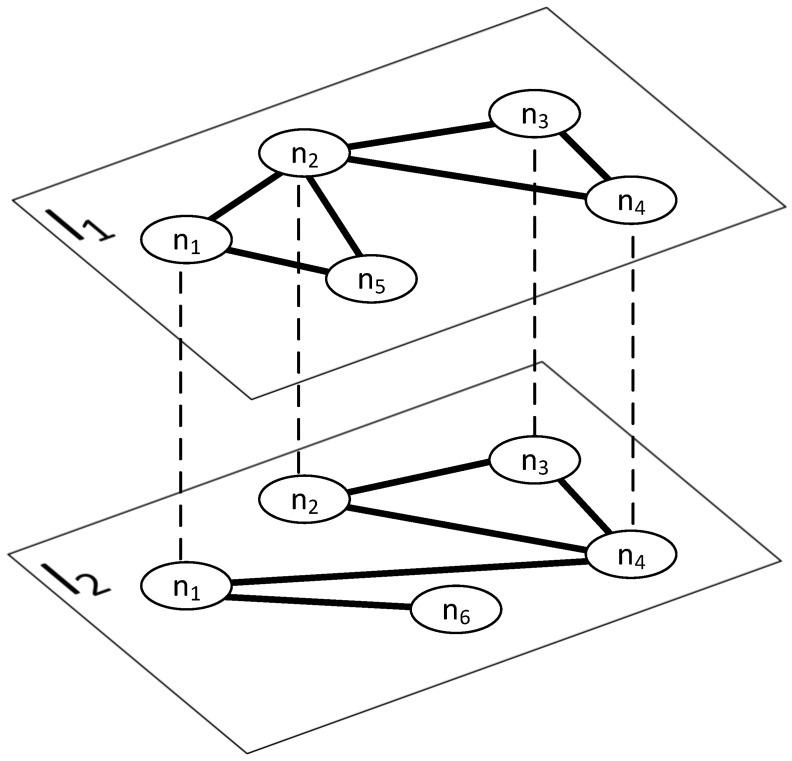
An example of multilayer networks.

**Figure 2 entropy-25-00231-f002:**
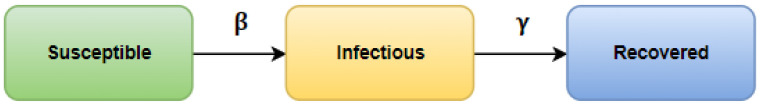
State changes in SIR model.

**Figure 3 entropy-25-00231-f003:**
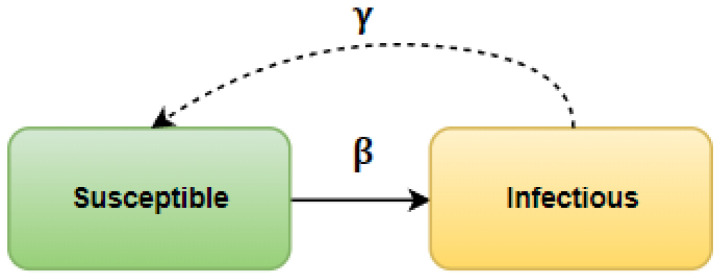
State changes for SIS model.

**Figure 4 entropy-25-00231-f004:**
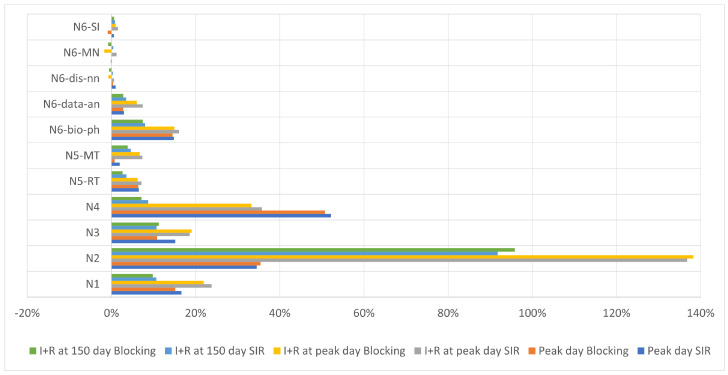
The results for different scenarios; our baseline is SIR and UAU to which we compare two other processes. Each bar represents how much faster the peak day was or how many more nodes became infected (until the peak day or until day 150) compared to SIR and UAU, that is, the scenario where both the virus and the information start to spread at the same time.

**Figure 5 entropy-25-00231-f005:**
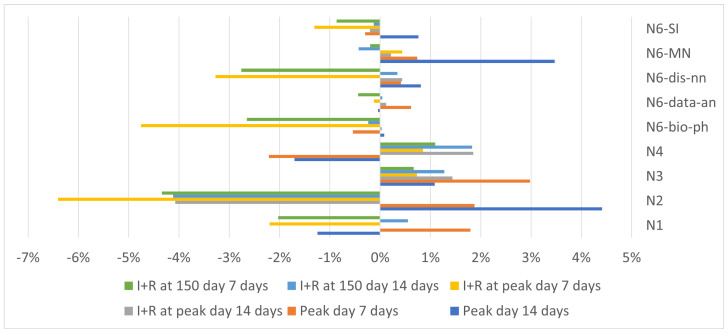
The results for different delay times (the baseline is SIR and UAU with 21 days delay) to which we compare two other delay periods. Each bar represents how much sooner or later (in case of negative values) the peak day was, or how many more or fewer (in the case of negative values) nodes became infected till peak day or till day 150.

**Table 1 entropy-25-00231-t001:** Summary of the experimental setup.

Param.	Values	Description
β	0.19, 0.22, 0.28, 0.31	The probability of contracting an infection during contact with an infected individual.
γ	0.1, 0.02, 0.08	The probability of recovery of an infected individual during each iteration.
$\beta^{\prime }$	$\frac{\beta }{10}$	The probability of contracting infection by aware person.
ϵ	min(β∗x,1);xε{1,2,3,4}	The probability of an unaware person contracting the information from their aware neighbor.
μ	min(γ∗x,1);xε{1,2,3,4}	The probability of an aware person forgetting the information or stopping being influenced by it.
$\epsilon^{\prime }$	0.692	The probability of an unaware infected person to become aware.
time	150	We simulated the first 150 days of the pandemic.
repetition	20	The simulation was repeated 20 times for each combination of parameters and each network.

**Table 2 entropy-25-00231-t002:** Probabilities change by spreading processes interactions.

State in SIR Model	State in UAU Model	Probability Change
Susceptible	Unaware	-
Infectious	Aware	-
Recovered	Unaware	-
Susceptible	Aware	$\beta -\beta^{\prime }$
Infectious	Unaware	$\epsilon -\epsilon^{\prime }$
Recovered	Aware	-

**Table 3 entropy-25-00231-t003:** Networks used in experiments, their parameters, and short description. The average degree of actors was calculated using degree definition from [43]. For each direct contact layer, the average node degree on that layer is included in the brackets.

Net.	Layers	Nodes	Edges	Avg. Degree	Direct Contact Layer	Description
N1	5	61	620	20.33	work (6.47)	AUCS CS-AARHUS [44]
N2	3	241	1370	11.37	advice (4.18)	Ckm Physicians Innovation [45]
N3	37	417	3588	17.21	Ryanair (9.39)	EU Air Transportation [46]
N4	3	71	1659	46.73	co-work (10.8)	Lazega Law Firm [47]
N5	3	88,804	210,250	4.64	RT (2.79) and MT (3.83)	Tweets related to 2013 World Championships in Athletics [48]
N6	13	14,489	59,026	8.39	physics.bio-ph (4.13), q-bio.MN (4.64), physics.data-an (5.30), cond-mat.dis-nn (4.19), cs.SI (4.69)	Arxiv papers related to Network Science [49]

**Table 4 entropy-25-00231-t004:** The results for different scenarios: our baseline is SIR and UAU to which we compare two other processes. The value in each cell represents how faster the peak day was or how many more nodes were infected (until the peak day or until 150 day) compared to SIR and UAU, that is, the scenario where both the virus and the information start to spread at the same time.

	Peak Day		I + R at Peak Day		I + R at 150 Day	
**Network**	** SIR **	**Blocking**	** SIR **	**Blocking**	** SIR **	**Blocking**
N1	16.65%	15.18%	23.78%	21.91%	10.70%	9.80%
N2	34.54%	35.42%	136.78%	138.23%	91.80%	95.83%
N3	15.16%	10.82%	18.60%	19.05%	10.73%	11.26%
N4	52.15%	50.74%	35.77%	33.28%	8.74%	7.14%
N5-RT	6.51%	6.36%	7.14%	6.24%	3.59%	2.67%
N5-MT	1.96%	0.78%	7.32%	6.75%	4.56%	3.88%
N6-bio-ph	14.85%	14.50%	16.03%	14.94%	8.00%	7.45%
N6-data-an	3.00%	2.81%	7.42%	6.00%	3.48%	2.78%
N6-dis-nn	1.02%	0.41%	0.65%	−0.68%	0.28%	−0.55%
N6-MN	−0.15%	0.02%	1.24%	−1.72%	0.44%	−0.82%
N6-SI	0.58%	−0.90%	1.57%	1.04%	0.87%	0.63%

**Table 5 entropy-25-00231-t005:** The results of the *p*-value for Wilcoxon signed rank test. We compare the results of SIR and blocking to SIR and UAU. Wilcoxson signed rank test is a nonparametric counterpart of the paired *t*-test and is often used in situations when we cannot ensure normal distribution of samples [50,51].

	Peak Day		I + R at Peak Day		I + R at 150 Day	
**Network**	** SIR **	**Blocking**	** SIR **	**Blocking**	** SIR **	**Blocking**
N1	<0.05	<0.05	<0.05	<0.05	<0.05	<0.05
N2	>0.05	<0.05	<0.05	<0.05	<0.05	<0.05
N3	<0.05	<0.05	<0.05	<0.05	<0.05	<0.05
N4	<0.05	<0.05	<0.05	<0.05	<0.05	<0.05
N5-RT	<0.05	<0.05	<0.05	<0.05	<0.05	<0.05
N5-MT	<0.05	>0.05	<0.05	<0.05	<0.05	<0.05
N6-bio-ph	<0.05	<0.05	<0.05	<0.05	<0.05	<0.05
N6-data-an	<0.05	<0.05	<0.05	<0.05	<0.05	<0.05
N6-dis-nn	<0.05	>0.05	<0.05	<0.05	>0.05	<0.05
N6-MN	>0.05	>0.05	>0.05	<0.05	>0.05	>0.05
N6-SI	>0.05	<0.05	<0.05	<0.05	<0.05	<0.05

**Table 6 entropy-25-00231-t006:** The results for different delay times (the baseline is SIR and UAU with 21 days delay)to which we compare two other delay periods. The value in each cell represents how much sooner or later (in case of negative values) the peak day was, or how many more or fewer (in the case of negative values) nodes became infected till peak day or till day 150.

	Peak Day		I + R at Peak Day		I + R at 150 Day	
**Network**	**14 days**	**7 days**	**14 days**	**7 days**	**14 days**	**7 days**
N1	−1.25%	1.80%	0.00%	−2.20%	0.55%	−2.03%
N2	4.41%	1.88%	−4.07%	−6.41%	−4.12%	−4.34%
N3	1.09%	2.98%	1.44%	0.74%	1.28%	0.67%
N4	−1.70%	−2.21%	1.85%	0.85%	1.83%	1.10%
N6-bio-ph	0.08%	−0.54%	0.03%	−4.76%	−0.24%	−2.65%
N6-data-an	−0.04%	0.62%	0.12%	−0.12%	0.05%	−0.44%
N6-dis-nn	0.81%	0.42%	0.44%	−3.27%	0.34%	−2.76%
N6-MN	3.47%	0.74%	0.21%	0.44%	−0.43%	−0.20%
N6-SI	0.76%	−0.30%	−0.20%	−1.31%	−0.13%	−0.86%

**Table 7 entropy-25-00231-t007:** The results of the *p*-value for Wilcoxon signed-rank test. We compare the results of SIR and UAU with 7 and 14 days’ delay to SIR and UAU with 21 days’ delay. Wilcoxon signed-rank test is a nonparametric counterpart of the paired *t*-test and is often used in situations when we cannot ensure normal distribution of samples [50,51].

	Peak Day		I + R at Peak Day		I + R at 150 Day	
**Network**	**14 days**	**7 days**	**14 days**	**7 days**	**14 days**	**7 days**
N1	>0.05	>0.05	>0.05	>0.05	>0.05	<0.05
N2	>0.05	<0.05	>0.05	>0.05	>0.05	>0.05
N3	>0.05	>0.05	>0.05	>0.05	>0.05	>0.05
N4	>0.05	>0.05	>0.05	>0.05	>0.05	>0.05
N6-bio-ph	>0.05	>0.05	<0.05	>0.05	>0.05	<0.05
N6-data-an	>0.05	>0.05	>0.05	>0.05	>0.05	<0.05
N6-dis-nn	>0.05	>0.05	<0.05	>0.05	>0.05	<0.05
N6-MN	>0.05	<0.05	>0.05	>0.05	>0.05	>0.05
N6-SI	>0.05	>0.05	>0.05	<0.05	>0.05	<0.05

## Data Availability

All datasets are available at CoMuNe Lab, Research Group for Multilayer Modeling and Analysis of Complex Systems https://manliodedomenico.com/data.php (accessed on 23 January 2023).

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
