# Peer review of "Influence of Information Blocking on the Spread of Virus in Multilayer Networks"

_entropy, 2023, doi:10.3390/e25020231_

Round 1
Reviewer 1 Report
The authors investigate the role of information blocking in epidemic dynamics on top of multilayer networks. The idea and insights are interesting, but it is not novel and the simple use of real network data did not convince me to consider this paper as interesting to the readers of Entropy. Some general sentences are also biased such as mentioning only European countries as those "where the epidemic took a drastic course", and using many journalistic references blaming China for the SARS-CoV-2 (Refs. 2-6). Also, please see the following works that are related to this one:- 10.1103/PhysRevE.106.034307
- 10.1016/j.physa.2021.126558 Some specific points:
- Introduction: "The development of human mobility patterns" -> is "development" the best word here?
- There are also some problems with the definitions of multilayer networks. What are $l_1$ and $l_2$ in line 33 of page 1? What are $n_1$ and $n_2$? - Does it make sense to use networks of retweets and mentions on Twitter to simulate a direct contact layer? And a network of co-authors on ArXiv? - It is not clear to me that the differences shown in Tables 4 and 5 are statistically relevant. Statistical tests are necessary to evaluate if the differences are due to the information blocking or the stochastic nature of the simulations.
Reviewer 2 Report
The topic covered by the article is very interesting. However, I would like to make the following comments to the authors:
1)The introduction should be modified, describing the following points:
* The problem to be addressed and the need to solve it
* Works related to this topic. The state of the art could be established here to solve this problem
* Brief description of the solution given by the authors.
* Next, give a description of the following manuscript's sections
2)ith respect a State Changes. In the model an actor changes their state
with probability beta, but also it is more probabily than an actor changes
their state if the number of neighbours with that state is higher. This
aspect should be considered by the authors. In the information spread, similarly, the dependence of an actor that changes its state on the number of neighbors with that new state can influence the strength of the change, but this aspect is not considered in the model. For example the probability of changing shouldn't be the same for an actor with two neighbour with the new state than other with seven neighbour
3).-What is the information of the authors to define \beta' as \beta/10. Is there a true ground about to choose this parameter in that way?
4) The section "Interaction between virus and informatin processes." is very interesting.
5) With respect to Table 4 it will be interesting to give the average degree
in the network.
6)A study interesting can be to choose the infected nodes initially as
those with high centrality. In this sense, different methods of centrality can be studied. From this approximation, discuss how the results are modified.
7) In the same way, the author can generate multilayer networks with strong communities and show the evolution of the different models.
Round 2
Reviewer 1 Report
The new references and statistical testing have improved the manuscript. However, I am still not convinced of the model's novelty and results, and there is no controlled study with synthetic networks with tuned parameters (such as average degree, heterogeneity, community structures, and so on) to better understand the results in different multilayer networks.
Reviewer 2 Report
I am glad about the revised manuscript.
